# An Update on African Swine Fever Virology

**DOI:** 10.3390/v11090864

**Published:** 2019-09-17

**Authors:** Axel Karger, Daniel Pérez-Núñez, Jesús Urquiza, Patricia Hinojar, Covadonga Alonso, Ferdinando B. Freitas, Yolanda Revilla, Marie-Frédérique Le Potier, Maria Montoya

**Affiliations:** 1Institute of Molecular Virology and Cell Biology, Friedrich Loeffler Institut, Federal Research Institute for Animal Health, Südufer 10, 17493 Greifswald-Insel Riems, Germany; axel.karger@fli.de; 2Virology Department, Centro Biología Molecular Severo Ochoa, CSIC-UAM, 28049 Madrid, Spain; daniel_perez@cbm.csic.es (D.P.-N.); yrevilla@cbm.csic.es (Y.R.); 3INIA, Instituto Nacional de Investigación y Tecnología Agraria y Alimentaria, 28040 Madrid, Spain; chuslucena12@gmail.com (J.U.); phinojar@gmail.com (P.H.); calonso@inia.es (C.A.); 4Centre for Interdisciplinary Research in Animal Health (CIISA), Faculty of Veterinary Medicine, University of Lisbon, 1649-004 Lisboa, Portugal; FerdinandoFreitas@fmv.ulisboa.pt; 5ANSES, Laboratoire de Ploufragan/Plouzané/Niort, Unité Virologie Immunologie Porcines, Anses, 22440 Ploufragan, France; marie-frederique.lepotier@anses.fr; 6Centro de Investigaciones Biológicas (CIB-CSIC), Ramiro de Maeztu 9, 28040 Madrid, Spain

**Keywords:** African swine fever virus, virology, proteomics, virus–host interaction

## Abstract

Animal diseases constitute a continuing threat to animal health, food safety, national economy, and the environment. Among those, African swine fever (ASF) is one of the most devastating viruses affecting pigs and wild suids due to the lack of vaccine or effective treatment. ASF is endemic in countries in sub-Saharan Africa, but since its introduction to the Caucasus region in 2007, a highly virulent strain of ASF virus (ASFV) has continued to circulate and spread into Eastern Europe and Russia, and most recently into Western Europe, China, and various countries of Southeast Asia. Given the importance of this disease, this review will highlight recent discoveries in basic virology with special focus on proteomic analysis, replication cycle, and some recent data on genes involved in cycle progression and viral–host interactions, such as I215L (E2 ubiquitin-conjugating enzyme), EP402R (CD2v), A104R (histone-like protein), QP509L, and Q706L (RNA helicases) or P1192R (Topoisomerase II). Taking into consideration the large DNA genome of ASFV and its complex interactions with the host, more studies and new approaches are to be taken to understand the basic virus–host interaction for ASFV. Proteomic studies are just paving the way for future research.

## 1. Introduction

Pork is the most widely consumed meat in the world, accounting for more than one-third of meat produced worldwide. Thus, infectious diseases affecting pigs threaten an important source of high-quality protein, and the globalization of the swine industry has contributed to the emergence and spread of pathogens. Among those pathogens, African swine fever virus (ASFV) has been, since its discovery in the early years of last century, one of the main concerns in pig industry [1]. ASFV is endemic in most sub-Saharan countries [2], but since its introduction in Georgia (2007), ASFV has spread quickly to other neighboring countries in Europe [3] and Asia. This is of particular concern in the case of China, producing half of the world’s pig population, where it was first reported in 2018 [4]. ASFV belongs to the family Asfarviridae, which includes the single species ASFV, isolates of which have linear dsDNA genomes of 170–194 kbp. Virions have an internal core, an internal lipid membrane, an icosahedral capsid, and an outer lipid envelope [5].

ASFV infection in domestic pigs and wild boar may have different outcomes depending on the viral strain, but it can result in the rapid death of most infected animals by an acute hemorrhagic fever with transmission by contact or ingestion, or by ticks of the genus Ornithodoros. Due to the lack of vaccine or effective treatments, options for controlling spread of this virus are limited to quarantine and culling of animals in infected farms. As African swine fever (ASF) is a notifiable disease to the World Organisation for Animal Health (OIE), up-to-date information on ASF outbreaks in domestic pigs and cases in wild boar is available on the OIE World Animal Health Information System (https://www.oie.int/wahis_2/public/wahid.php/Diseaseinformation/diseasehome). Also, the Food and Agriculture Organization (FAO) of the United Nations publishes updates on the current ASF situation (http://www.fao.org/ag/againfo/programmes/en/empres/ASF/situation_update.html).

Transmission of ASFV, its global distribution, and control strategies have been recently reviewed elsewhere [6]. Also, virus evasion strategies, as well as the new attempts to generate an effective vaccine, have been reviewed recently [7,8]. However, little attention has been focused onto the biology of the virus and its relation with cellular components during its viral cycle. Understanding the mechanisms by which the virus interacts with the cellular machinery in order to infect, replicate, and generate virus progeny is essential for developing future antiviral strategies. Therefore, this review will focus on the recent discoveries in ASFV virology and the gaps of knowledge in the field.

## 2. Proteomics on ASFV

One of the recent new approaches in understanding ASFV virology is to analyze ASFV-infected cells or virus particles using proteomic tools like two-dimensional gel electrophoresis, alone or in combination with mass spectrometry (MS). These have been applied in the past to analyze the expression of ASFV proteins or the cellular response to ASFV infection [9,10,11,12,13]. However, only recently have the proteome of the ASFV particle [14] and the intracellular ASFV proteome in cultured mammalian cells [15] been analyzed using modern MS platforms. This seems quite surprising, as MS analysis allows the sensitive and highly specific qualitative and quantitative analysis of proteins without the need for specific immunologic reagents, which are not usually available for the full complement of gene products of complex viruses like ASFV. Thus, for a significant proportion of the predicted ASFV open reading frames (ORFs), no information about the expression of the corresponding protein was available in the literature. This knowledge gap now has been partially closed.

In the study by Alejo et al. [14], the proteome of highly purified extracellular virions from the Vero-adapted strain BA71V was analyzed. Together with immunoelectron microscopic investigations, a comprehensive atlas of the ASFV particle was established. Of the 157 putative viral proteins encoded by BA71V, 68 were identified by MS. While essentially all previously known structural proteins could be confirmed, 44 so far unrecognized constituents of the virus particle were discovered, emphasizing the ‘open view’ character of MS. Functionally, structural proteins were mainly associated to virus structure and morphogenesis (16 proteins), viral transcription and RNA modification (13), maintenance of the genome integrity (4), virus entry (3), and host defence evasion (2). Surprisingly, 23 hitherto undetected proteins could be additionally identified which have no known functions but still represent 34% of the proteins present in the virus particle. Quantitative evaluation of the MS data confirmed the established main components of the virus particle (p72 and the core shell proteins derived from pp220 and pp62), but also revealed that some of the newly identified structural proteins are present in significant amounts, as, for example, pM1249L, pCP123L, pC129R, pC717R, and pI177L each accounted for over 1% of the virion protein content. Expectedly, a number of host proteins were found associated with the virus particle, mostly typical plasma membrane proteins or proteins associated with the cortical actin cytoskeleton.

In the study by Keßler et al. [15], the expression of ASFV genes was studied in three mammalian cell lines after infection with a recombinant derivative of the OURT88/3 strain. The cell lines (WSL-HP from wild boar, human HEK 293, and Vero cells from *Chlorocebus sabeaus*) were chosen as they derive from susceptible and non-susceptible host species. For 94 of the 157 ORFs present in OURT88/3, the corresponding protein was identified in one or more of the cell lines; for 23 of them, this was the first evidence for expression. Of the 57 ASFV proteins expressed in infected HEK 293, 54 were also present in both other lines, suggesting that these may be strictly required to maintain ASFV infection in cultured mammalian cells. In Vero and WSL-HP cells, 88 and 83 ASFV proteins were expressed, respectively, with an overlap of 78, indicating that the course of the infection in these cells may be similar but distinct from HEK 293 cells. This was also confirmed by the comparison of the expression levels of individual ASFV proteins, which correlated well between Vero and WSL-HP, while larger differences were observed in comparison to HEK 293 cells. Quantitative evaluation, surprisingly, also revealed that some of the most strongly expressed ASFV proteins are so far uncharacterized and may therefore be preferred candidates for functional characterization. Examples for such proteins are pK145R and pI73R, which both rank among the five most abundant viral proteins in all three cells.

The summarized results of the recent proteomic studies [14,15] in Figure 1 show that for most of the predicted ASFV ORFs, the existence of the corresponding protein in mammalian cell culture could be demonstrated using MS. However, evidence for expression still lacks for other ORFs, many of them members of the multigene families (MGFs), indicating that these may play a role only in infected animals or in infected primary macrophages, the natural host cells for ASFV replication, which have not been subject to proteome analysis so far.

## 3. Cell Entry of ASFV

The mechanism whereby any virus enters host cells might be crucial for inhibiting viral infection and constitutes a potential target for treatment or vaccination. However, this step has been—and still is—a controversial matter reviewed recently for ASFV [8,16]. Early studies described that ASFV entry is as fast as 30 min post when nearly 60% of virions have been internalized [17]. Surprisingly, the identity of the cellular receptor(s) for ASFV still remains elusive, as well as the viral ligand(s) of attachment. Nevertheless, there are new findings shedding light onto the steps the virus uses in its cellular replicative circle.

ASFV uses a complex entry process involving several cellular factors and mechanisms. In early studies, ASFV was described to enter host cells by receptor-mediated endocytosis, transiting along the endolysosomal pathway, and penetrating in a low pH-dependent manner after a presumed fusion event [17,18]. Later studies identified classical clathrin- and dynamin-dependent endocytosis as the primary entry route for ASFV [19,20]. Additionally, it has also been described that ASFV uptake by macropinocytosis is a nonselective endocytosis involving fluid-phase uptake, where the virus causes cytoplasm membrane perturbation, blebbing, and ruffles [21]. These different pieces of evidence have been related to the employment of complementary methodologies to assess virus uptake and to the usage of different target cell types and non-highly purified virus preparations [22].

ASFV entry into host cells depends on several factors, such as temperature, cholesterol, energy, and vacuolar pH [16,21]. In summary, ASFV has been reported to internalize via two endocytic pathways: Macropinocytosis and clathrin-mediated endocytosis (CME). ASFV enters the cell by clathrin- and dynamin-mediated endocytosis in Vero cells and pig macrophages [17,19,23]. Clathrin-coated vesicles mediate the fastest uptake mechanism at the plasma membrane and direct vesicle traffic to the endocytic pathway. ASFV entry occurs rapidly, in seconds, and viral particles can be detected as early as 1–15 min after infection inside endosomes [24]. Upon receptor activation, clathrin molecules are recruited to the cell surface, together with adaptor proteins epsins, AP-2, etc. Dynamin is another associated protein, a medium-size GTPase that mediates scission at the collar of the endocytic invagination, allowing the individualization or budding required for the release of the endocytic vesicle at the cytosolic side of the membrane. A hallmark protein endocytosed by CME is also transferrin. The inhibition of assembly of clathrin with inhibitors such as chlorpromazine or interfering RNA inhibiting adaptor proteins causes subsequent inhibition of clathrin-mediated endocytosis of hallmark transferrin, vesicular stomatitis virus (VSV), and ASFV entry [23]. Caveolae-mediated endocytosis (CAV) mediates slower internalization and is linked to receptors at the membrane lipid rafts used by SV40 or polyoma virus. The inhibition of dynamin with dynasore drug has a potent antiviral effect, blocking both CME and CAV, thereby inhibiting entry of several viruses.

Cortical actin rearrangements are required for the formation of the clathrin-coated vesicles [25], especially in large virus endocytosis (e.g., rabies virus and ASFV [26]). These are accompanied by Rac1 activation at an early stage of ASFV infection [27]. In fact, constitutive macropinocytosis of macrophages is important for ASFV entry [21,23], as shown by inhibition of ASFV entry by inhibitors of PAK1, Rac1, or the anion pump inhibitor EIPA-Amiloride [19,21,23]. Also, ASFV entry depends on cholesterol, and drugs inducing removal or disorganization of cholesterol at the plasma membrane are able to inhibit ASFV infection [19,28].

## 4. Endosomal Traffic of ASFV

Upon endocytosis, incoming ASFV particles move along the entire endolysosomal pathway, from peripheral early endocytic/macropinocytic vesicles, containing Rab5 and EEA1 markers, to perinuclear late endocytic compartments and lysosomes (Rab7+, CD63+, Lamp1+, cathepsin L+) (reviewed recently in [29]). All these routes of entry direct ASFV to the endosomal traffic where, in a few minutes, CME results in activation of signaling molecules of the early endosome (EE) [30]. Among them, PI3-kinase (PI3K) is an essential molecule that directs endosomal traffic and maturation. ASFV induces EGFR and PI3K-Akt pathway activation [21]. PI3K inhibition by wortmannin strongly inhibits early ASFV infection [24]. Since activation of Rac1-GTPase has been involved in the regulation of macropinocytosis by triggering membrane ruffling, it is not surprising that ASFV also triggers Rac1 activation to enter into the host cells [21]. Similarly, p21-activated kinase 1 (Pak1), a serine/threonine kinase, has a key role in ASFV infection as it is one of the most relevant kinases related to several virus entry processes since it is involved in the regulation of cytoskeleton dynamics and is needed during all stages of micropinocytosis [21].

Also, ASFV starts its traffic along the endocytic pathway at the EE. Entering virions can be detected from 1–15 min after virus addition in Vero cells by using antibodies against the major ASF viral capsid protein p72 [24]. Maturation within the EE would entail acidification of the endosomal lumen along late endosomal compartments and progressive changes in the surface expression markers of these vesicles. Under the harsh conditions of the endosomal lumen virion decapsidation as the first step of ASFV uncoating occurs [24], thereby the virion characteristic icosahedral morphology is no longer observed [23,31]. Bafilomycin inhibition of the v-ATPase, strongly inhibits ASFV infection, increasing the number of virions with intact capsids found in the endosomes [32]. However, decapsidated virions labeled with antibodies against core proteins are found in untreated endosomes at 30–45 min after infection, which is the time of uncoating.

After decapsidation, the next step would be fusion of the now exposed internal virion membrane with the endosomal limiting membrane. Interestingly, virus fusion is dependent on virus protein pE248R, a transmembrane polypeptide of the inner envelope [23]. Fusion is strongly dependent on cholesterol [31]. In fact, the intake of dietary cholesterol parallels virus entry. Dietary cholesterol enters the cell bound to low-density-lipoproteins (LDL) by the LDL-receptor. Cholesterol entry is a clathrin-mediated process and this lipid traffics the endosomal pathway and gets enriched at the intraluminal vesicles (ILVs) of the multivesicular bodies (MVBs), where it is transformed in free cholesterol. In order to reach its cellular distribution, cholesterol flux from the endosome is highly regulated by cholesterol transporters at the endosomal membrane. The integrity of this cholesterol transport and the proteasome function is required for a successful viral fusion and further replication [31,32]. In fact, ASFV infection redistributes the entire vesicular system to promote this process, and redirects endosomal traffic to the early replication site in an area close to the nucleus [33]. Thereby, high amounts of endosomal membranes are recruited and accumulate at early replication sites, providing the membrane support and cholesterol supplies required to establish a viral replication organelle known as viral factory (VF).

ASFV ensures an appropriate lipid flux to the VF, which is mediated by lipid transfer proteins that are located in the cytoplasm and show high affinity for oxysterols and cholesterol [34]. These sterol-shuttling proteins, such as oxysterol-binding protein (OSBP) and OSBP-related proteins (ORP), are abundant in membrane contact sites, which are essential to allow lipid and ion exchange between organelles. Furthermore, recent studies have shown their recruitment to the viral factory in ASFV-infected Vero and HeLa cells [34]. Although there are many functions that still remain unknown, it is certain that these proteins play a role in lipid sensing, transport, and also cell signaling [35,36]. Therefore, successful infection of ASFV relies partially on these proteins, as they mediate an efficient way to transfer lipids to the VF. Given the implication of these proteins in lipid flux, they have also been proven to be essential for replication of other viruses. In fact, inhibition of lipid transfer mediated by OSBP has been shown to be effective in inhibiting infection by some viruses, such as Enterovirus or Hepatitis C Virus [37].

## 5. CD2v Interaction with Adaptor Protein 1 (AP1)

CD2v (EP402R) is an ASFV protein with sequence homology to the T-lymphocyte surface adhesion receptor CD2. It possesses an extracellular N-terminal (Nt) region composed of two immunoglobulin-like domains, while the cytosolic C-terminal region (CD2v-Ct) shares no obvious amino acid sequence with the cellular CD2 cytoplasmic domain [38]. CD2v expression is required for the hemadsorption phenomenon observed in ASFV infected cells, which is most probably caused by its extracellular domain [38]. This may offer a mechanism for viral spread, since in pigs infected with a recombinant mutant ASFV Malawi lacking the CD2v protein (Malawi Δ8DR), both dissemination to lymph nodes and the onset of disease were delayed [39]. CD2v appears to be truncated in attenuated strains, suggesting the involvement of the protein in ASFV pathogenesis [40]. CD2v has also been reported to be involved in replication in the tick vector [41], and when ectopically expressed, the CD2v-Ct domain seems to be responsible for its Golgi location [42] and for binding to the actin-binding cellular protein SH3P7 [43].

Adaptor protein 1 (AP-1) is a cytosolic heterotetramer involved in the transfer of protein cargo from the trans-Golgi network (TGN) to endosomes [44]. AP-1 recruits clathrin to form clathrin-coated vesicles and plays a key role in selecting the cargo by distinguishing sorting signals in the cytoplasmic tail of integral membrane proteins. To date, two sorting signals have been identified and well characterized: The tyrosine (YXXF) and the di-leucine ([D/E]XXXL[L/I]) motifs [45], selectively recognized by AP-1. Recruitment of AP-1 to the TGN membrane is regulated by the small GTPase ADP-ribosylation factor 1 (Arf1) which is regulated by Golgi-associated GTPase-activating proteins (GAPs) and guanine nucleotide exchange factors (GEFs) [46]. GEF families (such as GBF1, BIG1/BIG2) are sensitive to brefeldin A (BFA) [47]. BFA inhibition of GEFs triggers the release of Arf1 from the Golgi membranes and consequently of AP-1 [48]. Therefore, binding to AP-1 might impact reorganization of the Golgi and cellular traffic events during ASFV infection, most likely facilitating viral replication, encapsulation, and/or egress. This viral strategy appears to be a common mechanism to increase virulence and disease progression observed in other viruses. For example, human immunodeficiency virus (HIV) Nef binds to AP-1 through a well characterized di-Leucine (di-Leu) motif [49,50]. This binding leads to the stabilization of AP-1on the membranes [51] and results in alteration of the endocytic pathway, correlating with an increase in virulence [52]. E6 protein from bovine papillomavirus type 1 (BPV-1) binds to AP-1 [53], while the interaction between AP-1 and glycosylated proteins of herpes simplex virus is involved in viral spread [54,55]. Interaction of AP-1 with viral proteins is proposed to be a mechanism to subvert host cellular trafficking to ensure and enhance viral morphogenesis and viral exit, whereas putatively allowing immune evasion of the infected cell [52,53,56].

CD2v binding to AP-1 and localization around the viral factory during ASFV infection was shown for the first time by Pérez-Núñez et al. [57]. These authors also demonstrated that binding was sensitive to BFA, and, as a consequence, AP-1 was released from the TGN membrane during the infection. Thus, in ASFV infected cells, AP-1 was dispersed into the cytoplasm, whereas CD2v remained attached to the membrane around the viral factory. Sequence analysis of CD2v with the ELM resource [58] identified a di-Leu motif predicted to mediate binding to AP-1. This di-Leu motif was indeed functional in the HIV protein Nef, but, unexpectedly, the motif was not involved either in the co-localization or in the interaction between CD2v and AP-1. Finally, a region of the cytoplasmic tail of CD2v that does not contain the di-Leu motif was identified. This region interacts with AP-1 in a pull-down assay, indicating that this region might anchor an as yet uncharacterized AP-1 binding motif. Interestingly, the clathrin adaptor complex AP-1 co-localization with CD2v around the viral factory was observed (Figure 2), thus suggesting protein–protein interaction. Together, these results demonstrate an important function for CD2v in AP-1 binding and location that would have direct consequences for traffic remodeling and virus infectivity, describing a new motif as being responsible for the interaction between viral protein and AP-1.

## 6. ASFV Genes Involved in Cycle Progression and Viral-Host Interactions

The genome of ASFV varies between 170 and 194 kb in length. These gross differences in genome size are predominately due to differences in the copy number of five different multigene families (MGFs). A recent review has explored the host and viral genetics contribution to pathogenesis and the different disease outcomes seen in different hosts [59]. However, there are some recent results about genes that are particularly involved in the virus cycle and host interaction which require further attention.

## 7. I215L—E2 Ubiquitin-Conjugating Enzyme

The identification of ASFV E2 ubiquitin-conjugating enzyme, ORF I215L [60,61], suggests that cellular ubiquitination pathway could be subverted during the ASFV infection, like described for other viruses [62,63,64,65]. More recently, I215L characterization showed that this putative viral E2 enzyme (ORF I215L) truly acts like an ubiquitin-conjugating enzyme, being mono and di-ubiquitinated. It was also pointed out the ability of this enzyme to remain catalytically active under a wide range of pH values (4 to 9) [66], which may be critical during the cell entry process, known to occur via a low-pH-dependent endosomal pathway [23,24,26], and also in the midgut epithelial cells of *Ornithodoros* spp. ticks, where the pH levels are lower than 7 [67]. This catalytic plasticity was also revealed under a broad range of temperatures (4 to 42 °C), as this can be important for the virus to remain active during the high fever episodes found in the infected animals, but also in the vector usually exposed to ambient temperature oscillations. Moreover, mono-, di-, and poly-ubiquitinated species were identified with detergent-soluble protein fractions extracted from infected cells, suggesting that pI215L may participate in distinct regulation mechanisms, since the ability to generate diverse substrate-ubiquitin structures is essential to target different host/viral proteins [66]. I215L viral gene is transcribed from early infection times, showing two main transcription peaks (at 2 and 16 hpi), suggesting that pI215L may be involved in distinct phases of the viral life cycle (e.g., viral transcription, genome replication, and viral egress) [66], as reported for several viruses [68]. Additionally, the detection of pI215L from 4 to 20 hpi and immunolocalization studies revealed that pI215L is recruited to viral factories, supporting the hypothesis that pI215L is involved in RNA transcription and/or DNA replication. Furthermore, the diffuse distribution of pI215L throughout the cytoplasm and nucleus may be considered related to the role in ubiquitination of viral proteins and/or host proteins involved in other functions (e.g., antiviral responses, DNA damage responses). Finally, results from siRNA experiments clarified that pI215L is involved in the late viral transcription as I215L downregulation lead to the reduction of the number of B646L transcripts, a decreased number of ASFV genomes (between 63 to 68%), and a reduced viral progeny (up to −94%). These new insights suggest that ASFV genome replication, viral late transcription, and progeny production are mediated through the ubiquitin pathway [66], corroborating previous studies showing the importance of the ubiquitin–proteasome system during the infection [32].

## 8. A104R—Histone-Like Protein

The ASFV genome encodes for ORF A104R, a putative histone-like protein that shares about 25% sequence identity with bacterial histone-like proteins (HU and IHF) [69,70].

Recent studies showed that purified recombinant pA104R binds dsDNA with higher affinity than ssDNA, suggesting that this protein is more efficient at folding full-length ASFV genomes rather than intermediate single-stranded genomes. Furthermore, in vitro studies showed that pA104R DNA-binding activity is maintained under a wide range of temperatures (4 to 37 °C) and pH values (4 to 11), probably to support ASFV replication in different hosts (soft tick vector and swine). Characterization studies revealed that pA104R has an optimal binding site size of around 14 to 16 nt and a minimal binding length of 11 to 20 nt [71], similar to other viral DNA-binding proteins [72,73]. Furthermore, pA104R has the capability to supercoil DNA in the presence of ASFV topoisomerase II [74,75]. This activity is described in bacterial histone-like proteins [76,77] and also in some viral proteins involved in genome packaging [78,79], suggesting that pA104R may be involved in ASFV genome packaging, which is supported by the distribution of pA104R over the central nucleoid structure [69,80]. The late transcription of A104R gene is corroborated by pA104R expression from 12 hpi onward, but not in the presence of Arabinose AraC, a strong transcription inhibitor. The recruitment of pA104R to viral factories strengthens the idea that this viral protein may participate in ASFV genome packaging [71], whereas, its nuclear localization suggests that pA104R may also participate in the viral DNA replication and/or in the heterochromatization of the cell genome [81], thus facilitating the viral infection—although this is not its main function. Finally, the A104R downregulation by siRNA (−27% at 16 hpi) has been proven to affect the production of infectious progeny given by a reduction in the viral yields (−82.0%), less viral genome copy number (−78.3%), and late viral transcription repression (B646; −47.6%) still, without interfering with the transcription of the early viral gene CP204L [71].

## 9. QP509L and Q706L RNA Helicases

ASFV genomes encodes about 20 genes that are considered to be involved in the transcription [82]. Besides several other critical putative enzymes, ASFV encodes two putative RNA helicases, the QP509L (58.103 kDa) and Q706L (80.376 kDa) [83,84]. These types of enzymes are known to be involved in DNA–RNA and RNA–protein interactions that occur from the beginning of viral gene expression and culminate with the release of infectious particles [85,86]. Recent studies showed that QP509L and Q706L ASFV RNA helicases are highly conserved among virulent and non-virulent isolates and cluster with other SF2 RNA helicases encoded by NCLDV [87], corroborating previous studies [82,88,89,90,91]. Regarding the transcription dynamics of the two ASFV SF2 RNA helicases, maximum mRNA levels were detected between 8 and 12 hpi, suggesting that both enzymes are mainly required during the intermediate and late stages of the infection cycle, when the viral DNA replication and transcription are more active. In fact, pQP509L was detected from 12 hpi within viral factories and host nucleus, whereas pQ706L was detected only at viral factories from 12 hpi onwards, indicating that both ASFV RNA helicases have different roles during replication cycle. Despite an early intranuclear phase proposed for ASFV [81,92], the presence of pQP509L in this cellular compartment, at later times of infection, can be related to other viral events than transcription and/or DNA replication as, for example, modulation of host proteins like described for RNA helicases of other viruses [93,94,95]. siRNA assays showed that QP509L and Q706L depletion induces a decrease of the late viral transcripts (ASFV-B646L), a reduced number of viral genomes coupled with a decreased viral yield, suggesting that both ASFV RNA helicases have relevant and non-redundant functions, not rescued by cellular RNA helicases [87]. A similar situation is found in other viral SF2 DEAD-box RNA helicases [96,97,98,99,100].

## 10. P1192R—Topoisomerase II

ASFV is the only known virus infecting mammals which encodes its own topoisomerase II (ORF P1192R) [101]. Type II topoisomerases regulate the DNA topology during replication, transcription, chromosome condensation–decondensation, and segregation by catalyzing transient double-stranded breaks in one helix DNA [102,103,104]. Previous studies showed that fluoroquinolones induce a viral genome fragmentation and delay of the viral protein synthesis [105]. Follow up studies also showed that ASFV-infected cells exposed to enrofloxacin during the late phase of infection (from 15 to 16 hpi) present induced-fragmentation of viral genomes, whereas no viral genomes were detected whenever enrofloxacin was added from 2 to 16 hpi, corroborating previous data [75]. These results suggest that enrofloxacin interferes with DNA resealing at the DNA cleavage, just like found in prokaryotes. Later studies using temperature-sensitive *Saccharomyces cerevisiae* demonstrated the functionality of P1192R to relax supercoiled DNA through complementation and in vitro decatenation assays, further confirmed by mutating its predicted catalytic residue [74,106]. Further characterization revealed that ASFV P1192R gene is actively transcribed as early as 2 hpi, reaching a maximum peak of accumulation around 16 hpi [75], and pP1192R co-localization in the viral factories at intermediate and late phases of infection when viral DNA synthesis, transcription, and translation are more active [106]. Following these assays, siRNA knockdown experiments revealed that ASFV Topoisomerase II plays a critical role in viral DNA replication and gene expression, with transfected cells displaying a decreased number of viral transcripts combined with reduced cytopathic effect when compared to the infected control group. Moreover, a significant decrease in the number of both infected cells and viral factories per cell and of the virus yields was also detected [75]. These data highlight the importance of P1192R during ASFV infection, opening a path to the use of specific drugs against this viral infection or an opportunity to generate live-attenuated virus in order to control it [81,82,83,84,85,86,87,88,89,90,91,92,93,94,95,96,97,98,99,100].

## 11. Summary

Since the first description of ASF in the early twentieth century, ASFV has been circulating in different parts of the world, where it is endemic (Africa) or has made recurrent outbreaks in other territories. Research in ASFV has followed these trends according to every outbreak, reaching a low of publications just before the recent outbreak in 2007. This review has highlighted recent discoveries in basic virology, with special focus on proteomic analysis, replication cycle, and some recent data on genes involved in cycle progression and viral-host interactions, such as EP402R (CD2v), I215L (E2 ubiquitin-conjugating enzyme), A104R (histone-like protein), QP509L, Q706L (RNA helicases), and P1192R (Topoisomerase II). Considering the dimension of this virus and the fact that its genome may vary between 170 and 194 kb in length, more efforts should be dedicated to basic virology research in each individual ORF, as well as the whole virus, so we can understand the mechanisms utilized by the virus in its replication cycle and its complex interaction with the host. Proteomic studies have just paved the way for future research.

## Figures and Tables

**Figure 1 viruses-11-00864-f001:**
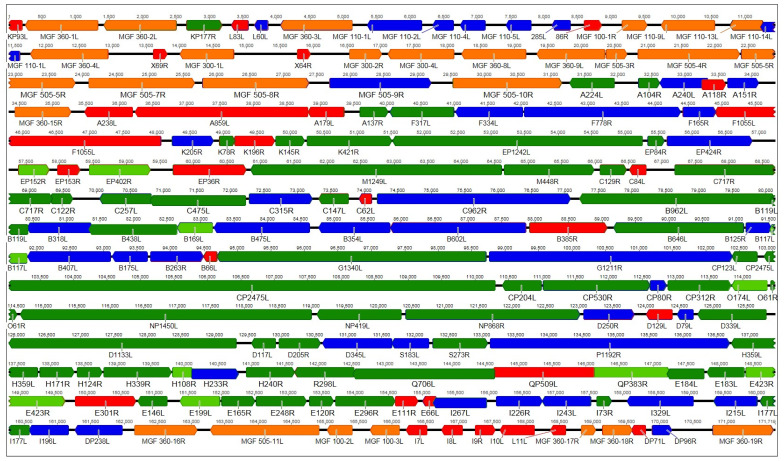
Representation of the genome of African swine fever virus (ASFV) strain OURT88/3. Predicted ORFs are color-coded according to the results of the MS analysis by Alejo et al. and Keßler et al. [14,15]. Orange and red: Multigene family (MGF) members or other open reading frames (ORFs) not identified in either study, respectively. Green: Structural proteins identified in both studies (dark) or by Alejo et al. only (light). Blue: Intracellular proteins identified by Keßler et al. only. As different virus strains were used in the studies, results from Alejo et al. were transferred to the homologous genes in OURT88/3.

**Figure 2 viruses-11-00864-f002:**
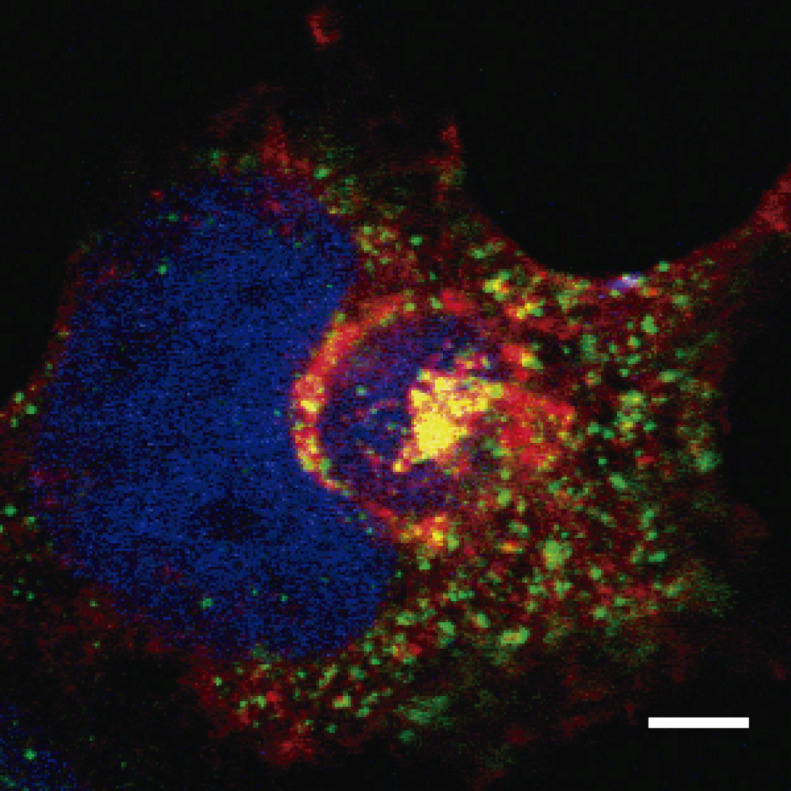
COS cells were infected with ASFV-E70. At 16 hpi, cells were fixed and immuno-stained with α-γ-adaptin (AP-1) (green) and α-CD2v (red). Cellular nucleus and viral factory were stained with DAPI (blue). Scale bar represents 5 µm.

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
