# Peer review of "An Update on African Swine Fever Virology"

_viruses, 2019, doi:10.3390/v11090864_

Round 1
Reviewer 1 Report
This review by Karger et al. on African swine fever (ASF) is timely and focuses on less known aspects of ASFV viral-host interactions. Therefore the manuscript merits publication, but I recommend the following modifications:
1) The quality of figure 1 needs to be improved. It is not possible to see all the ORFs.
2) Lines 108- 110. The authors should discuss the fact that the proteomic data was obtained in cell lines and not primary porcine macrophages. Therefore the statement that the MGFs or other ORFs that were not identified in these studies may play a role only in infected animals is pure speculation. They may indeed play an important role in primary macrophages, the natural host cells for ASFV replication.
3) Lines 219-221. The sentence is redundant and should be removed.
4) Lines 251-254. The sentence starting with “AP-1 has been shown” and finishing with “infected cell” would be better combined with the last sentence of the previous paragraph where the authors discuss the importance of AP-1 in other viral infections.
5) Line 287. Please avoid the use of the word nuclear if not referring to the cell nucleus, as this can be confusing. Perhaps crucial instead?
6) Lines 313 and 334. The authors speculate that both the histone like protein and the RNA helicases may be involved in immune modulation by modifying the host genome. However they should emphasize that their presence in the nucleus at late times post infection indicates that probably this is not their most important function, since inhibitors of host immune response are generally expressed early.
7) Line 316. Should read B646L.
8) Line 346. Please write “temperature sensitive saccharomyces cerevisiae” instead of “saccharomyces cerevisiae temperature sensitive”
9) Lines 349-353. The sentence starting with “Follow up studies” and finishing with “prokaryotes” should be moved to line 345 after “protein synthesis”. In addition, the general mechanism of action of fluoroquinolones should be explained. The paragraph would read better this way, moving from the general effect of these drugs on ASFV infection to the specifics of the topoisomerase gene.
Author Response
Review 1
English language and style are fine/minor spell check required
This review by Karger et al. on African swine fever (ASF) is timely and focuses on less known aspects of ASFV viral-host interactions. Therefore, the manuscript merits publication, but I recommend the following modifications:
1) The quality of figure 1 needs to be improved. It is not possible to see all the ORFs.
We have submitted a high-resolution copy of figure 1 as requested.
2) Lines 108- 110. The authors should discuss the fact that the proteomic data was obtained in cell lines and not primary porcine macrophages. Therefore, the statement that the MGFs or other ORFs that were not identified in these studies may play a role only in infected animals is pure speculation. They may indeed play an important role in primary macrophages, the natural host cells for ASFV replication.
The reviewer is right; a sentence has been added in the last paragraph on Proteomics on ASFV to clarify this point.
3) Lines 219-221. The sentence is redundant and should be removed.
This sentence has been removed from the revised version as suggested.
4) Lines 251-254. The sentence starting with “AP-1 has been shown” and finishing with “infected cell” would be better combined with the last sentence of the previous paragraph where the authors discuss the importance of AP-1 in other viral infections.
Following the reviewer´s comment, this sentence has been removed and a new one has been added at the end of the previous paragraph including this information.
5) Line 287. Please avoid the use of the word nuclear if not referring to the cell nucleus, as this can be confusing. Perhaps crucial instead?
We thank the reviewer for drawing out attention to this point. In the revised version, the word nuclear is now used only when referring to nuclear localization to avoid any confusion.
6) Lines 313 and 334. The authors speculate that both the histone like protein and the RNA helicases may be involved in immune modulation by modifying the host genome. However, they should emphasize that their presence in the nucleus at late times post infection indicates that probably this is not their most important function, since inhibitors of host immune response are generally expressed early.
As suggested by the reviewer, both sentences (for A104R and RNA helicases, QP509L and Q706L) were changed accordingly, in order to clarify the point and emphasize the secondary role in terms of antiviral modulation at later times of infection.
7) Line 316. Should read B646L.
This mistake was corrected as suggested.
8) Line 346. Please write “temperature sensitive saccharomyces cerevisiae” instead of “saccharomyces cerevisiae temperature sensitive”
The word order has been corrected.
9) Lines 349-353. The sentence starting with “Follow up studies” and finishing with “prokaryotes” should be moved to line 345 after “protein synthesis”.
Following the reviewer´s comment, we have changed this item.
In addition, the general mechanism of action of fluoroquinolones should be explained. The paragraph would read better this way, moving from the general effect of these drugs on ASFV infection to the specifics of the topoisomerase gene.
A short description explaining the mode of action of the fluoroquinolones was included as requested.
Reviewer 2 Report
In the review by Karger et al. the authors give a nice short overview of ASFV proteomics, cell entry and endosomal traffic of ASFV. The authors then decide to focus on a subset of ASFV proteins involved in cycle/progression and viral-host interactions. The work is well written, easy to read and the information togheter is useful. It highlights there are still significant gaps in ASFV virus-host interaction and proteomics is an important tool to understand basic ASFV virology.
I think that the article can be accepted in the present form. There are just three little points I would like to suggest:
- Please add [14] and [15] also in Figure Legend 1.
- In cell entry of ASFV, please can you some information on the time required for ASFV to enter in the cells?
- I think that chapter 6 should became chapter 5 and chapter 5, 7, 8, 9, 10 should became 5.1, 5.2, 5.3, 5.4, 5.5 respectively.
Author Response
Review 2
In the review by Karger et al. the authors give a nice short overview of ASFV proteomics, cell entry and endosomal traffic of ASFV. The authors then decide to focus on a subset of ASFV proteins involved in cycle/progression and viral-host interactions. The work is well written, easy to read and the information togheter is useful. It highlights there are still significant gaps in ASFV virus-host interaction and proteomics is an important tool to understand basic ASFV virology.
I think that the article can be accepted in the present form. There are just three little points I would like to suggest:
- Please add [14] and [15] also in Figure Legend 1.
Both references were added in Figure 1 legend as suggested by the reviewer.
- In cell entry of ASFV, please can you some information on the time required for ASFV to enter in the cells?
Following the reviewer´s comment, a number of sentences were added referring to the time required for ASFV to enter cells in the Cell entry section.
- I think that chapter 6 should became chapter 5 and chapter 5, 7, 8, 9, 10 should became 5.1, 5.2, 5.3, 5.4, 5.5 respectively.
The chapters were obviously renumbered by the journal during the submission process without us having any influence on that. The editor will be able to modify this point.